# Frontiers in the Standardization of the Plant Platform for High Scale Production of Vaccines

**DOI:** 10.3390/plants10091828

**Published:** 2021-09-02

**Authors:** Francesco Citiulo, Cristina Crosatti, Luigi Cattivelli, Chiara Biselli

**Affiliations:** 1GSK Vaccines Institute for Global Health, Via Fiorentina 1, 53100 Siena, Italy; francesco.x.citiulo@gsk.com; 2Council for Agricultural Research and Economics, Research Centre for Genomics and Bioinformatics, Via San Protaso 302, 29017 Fiorenzuola d’Arda, Italy; cristina.crosatti@crea.gov.it (C.C.); luigi.cattivelli@crea.gov.it (L.C.); 3Council for Agricultural Research and Economics, Research Centre for Viticulture and Enology, Viale Santa Margherita 80, 52100 Arezzo, Italy

**Keywords:** vaccines, plant factory system, virus-like particles, genome editing

## Abstract

The recent COVID-19 pandemic has highlighted the value of technologies that allow a fast setup and production of biopharmaceuticals in emergency situations. The plant factory system can provide a fast response to epidemics/pandemics. Thanks to their scalability and genome plasticity, plants represent advantageous platforms to produce vaccines. Plant systems imply less complicated production processes and quality controls with respect to mammalian and bacterial cells. The expression of vaccines in plants is based on transient or stable transformation systems and the recent progresses in genome editing techniques, based on the CRISPR/Cas method, allow the manipulation of DNA in an efficient, fast, and easy way by introducing specific modifications in specific sites of a genome. Nonetheless, CRISPR/Cas is far away from being fully exploited for vaccine expression in plants. In this review, an overview of the potential conjugation of the renewed vaccine technologies (i.e., virus-like particles—VLPs, and industrialization of the production process) with genome editing to produce vaccines in plants is reported, illustrating the potential advantages in the standardization of the plant platforms, with the overtaking of constancy of large-scale production challenges, facilitating regulatory requirements and expediting the release and commercialization of the vaccine products of genome edited plants.

## 1. Introduction

The advances in molecular biology techniques, together with the reverse vaccinology applications, have led to the development of recombinant subunit vaccines, which, unlike attenuated pathogens, are based on antigenic epitopes or sugar/protein/protein complexes [1,2].

For decades, recombinant protein vaccines have been produced at industrial scale in systems based on bacterial, insect, mammalian, and yeast cells that, however useful, require high costs and efficient equipment for large-scale fermentation and purification. Despite a modest presence of plant-expressed biopharmaceuticals on the market, the plant biopharming system (plant factory system) has been shown to be an effective biologic production host, with the full capacity to generate correctly folded and glycosylated therapeutic molecules [3,4]. It also offers a potential solution to the scalability and cost-effectiveness of large-scale production of vaccines. Although plant-based human vaccines and monoclonals are not approved for the market yet, several vaccine candidates against bacteria, fungi, or viruses have been successfully produced in various plant systems and tested in preclinical models, for immunogenicity and safety, or under clinical trials, for safety and efficacy [4,5,6].

In 2001, the US Defense Advanced Research Projects Agency sponsored the Blue Angel Project aimed at addressing the insufficient capability of providing vaccines against pandemics caused by new pathogens/strains and intentional biothreats. Within this project, a high containment, self-sufficient plant-based pharmaceutical production facility, able to manufacture 10 million doses of an H1N1 influenza vaccine in a single month, was created [7]. Plant-based production systems have been proposed to have the production pace that would be required to quell an unexpected viral outbreak, as in the case of the recent COVID-19 pandemic [8].

Plants, to some extent, can be transformed by multiple genes [9], and this makes relatively easy the in planta production of monoclonal antibodies resulting from genes encoding for the heavy and the light chain peptides, respectively. With the proper cis regulatory elements in the expression cassette, it is possible to quantitatively control the recombinant protein expression [10] or to address the production to specific tissues or organs. For example, seeds have been utilized as the first bioreactors for commercialized recombinant proteins because they offer the possibility to accumulate and store high amounts of target proteins [11].

The recent progresses in genome modification have brought the development of the targeted genome editing based on the CRISPR (Cluster Regularly Interspersed Short Palindromic Repeats)-Cas9 technique that allows an efficient and fast introduction of specific targeted mutations/deletions/insertions in specific sites of a genome [12,13]. This method has been successfully applied to plant genetic engineering, expediting the release of genome edited plants [14]. Nonetheless, to date, there are no vaccine candidates expressed in plants using the CRISPR-Cas9 approach. An example of its application in plant biofactoring is the inhibition of the plant glycosylation pathway to allow the production of non-glycosylated biomolecules [15,16]. Recently, CRISPR-Cas9 has been proposed as a tool for plant biomanufacturing to improve the expression of recombinant proteins in plant hosts and to eliminate host contaminants [17].

This review provides an overview of the most renewed techniques of vaccine development in plants and the synergism resulting from the conjugation of these technologies with genome editing methods based on CRISPR/Cas. The potential innovations in the standardization of the plant platforms are illustrated together with the overtaking of constancy of large-scale production challenges, thus expediting the release and commercialization of the vaccine products from genome edited plants. 

## 2. Transformation Technologies and CRISPR-Cas Genome Editing Methods

The expression of recombinant proteins and vaccines in plants can be achieved by introducing the corresponding DNA in the host cells through in vivo or in vitro plant transformation systems [18,19,20]. The most used transformation method is based on transient expression of recombinant DNA by different techniques: *Agrobacterium*-mediated transformation, delivery of “naked” DNA by particle bombardment, infection with modified viral vectors, polyethylene glycol (PEG)-mediated gene transfer, or electroporation of protoplasts [21,22,23,24,25,26]. Alternatively, plants can be stably transformed by *Agrobacterium* [27].

Transient transformation systems provide rapid expression, flexibility in gene stacking, capability to produce complex proteins and protein assemblies, and speed of scalability [26]. Moreover, they can be used to selectively transform non-reproductive organs, avoiding undesired spread events [24]. Currently, agroinfiltration and infection with modified viral vectors are the most frequently used transient expression methods [8]. Large-scale industrial production has been reached using systems based on the tobacco mosaic virus RNA replicon or derived from the bean yellow dwarf virus DNA replicon [28,29]. However, transient expression has some limitations that are mainly related to the high costs for bacteria or viral vector manipulation and production, and to the necessity of repeating the transformation in each cycle of production as the expression cassette is lost through generations [30].

Plant stable transformation allows the development of genetically modified (GM)/transgenic plants with well-defined master and working seed banks to be used for consecutive batches, leading to large-scale production [31,32]. Furthermore, cross-fertilization between transgenic plants producing two different recombinant proteins can generate siblings that express multiple recombinant genes, an approach useful for monoclonal antibodies purification [33]. For these reasons, transgenic plants have been depicted as the most suitable plant-based format for vaccine production [30].

The most utilized system for plant stable transformation is based on *Agrobacterium*. Nonetheless, the associated random insertion of the transgenes leads to significant inter-transformant variation in gene expression with transformants characterized by a low level of transgene expression. This is often caused by the integration of the exogenous DNA in genomic regions with low transcriptional activity, epigenetic control, or sequence-specific gene silencing events [17,34].

The genome editing method, based on CRISPR-Cas9, provides a simple, highly efficient, and versatile approach for the generation of site-specific mutations as well as site- specific insertion/deletion [14]. It is based on the activity of DNA-specific nucleases (typically Cas9) that cut DNA at 3 bases upstream of a 3–5 bp long sequence called Protospacer Adjacent Motif (PAM), leaving a blunt-end DNA double stranded break. The breaks are mainly repaired by the nonhomologous end-joining repair (NHEJ) mechanism that cuts the ends together, while the nearby sequences may be repaired by the homologous-derived repair (HDR) system [13]. Cas9 is recruited on a specific locus by a single guide RNA (sgRNA) complementary to a target sequence located upstream of PAM, where the cleavage occurs. The proper identification of the target site and the appropriate design of the sgRNAs are fundamental for the accurate and efficient targeting of the CRISPR/Cas9 system [14]. The method has been widely applied for targeted mutagenesis, gene knock-out, and for multiplexed gene editing in different plant species, producing high quality and sustainable products [35]. The presence of off-targets is limited as the nucleases are precisely directed towards specific sites by the sgRNAs. Furthermore, a self-cleaving system was developed and consisted of a transformation cassette including a constitutive promoter for Cas9 and sgRNA and an inducible promoter for an additional Cas9-specific sgRNA. This system leads to a user control over the duration of the cellular exposition to the Cas9 effect, minimizing the occurrence of off-targets. The additional sgRNA might be also directed towards the ends of the whole expression cassette allowing the production of transgene-free T0 plants carrying only the desired mutation/mutations [14]. 

Figure 1 illustrates the advantages of CRISPR-Cas9 with respect to *Agrobacterium*-mediated transformation.

The availability of deep annotated whole plant genomes in public databases and the user-friendly bioinformatic software and web-portals for the in silico design of sgRNAs and off-target searching tools (i.e., Cas-Designer, CasOT, COSMID, CRISPR-PLANT, CRISPR-P, GGGenome program, CRISPRdirect, CRISPR Genome Analysis Tool,) allow a precise construction of sgRNAs to be used for a CRISPR-Cas9-based gene targeting with minimal off-target effects [36,37,38].

In addition to gene knock-out, site-specific mutation, and regulation of gene expression, CRISPR/Cas9 has been utilized for the insertion or replacement of a single gene, multiple genes, or part of it/them (i.e., promoters or regulatory sequences) within a genome. To this aim, one or two double strand breaks are induced in the target sequence and the exogenous DNA, carrying arm sequences homolog to the ones flanking the cleavage site/sites, and is inserted into the cleaved site by HDR [14,39]. Examples have been reported in plants even if knock-in events by CRISPR/Cas9 in higher eukaryotes occur at very low rates because the double strand breaks are mainly repaired by NHEJ (on average 30–70%). For this reason, strategies to enhance the knock-in efficiency are under development and are based on the inhibition of NHEJ or stimulation of HDR, the design of a proper donor template, and the use of single stranded DNA donors with neutral substitutions in the recognition sites of sgRNAs and in PAMs to prevent subsequent excision events [39,40].

Despite these efforts, due to the lower frequency of HDR in plants, genome instability, and unpredictable outcomes of DNA repair, targeted gene/allele replacement through genome editing remains very challenging. Recently, two new precise and efficient genome editing systems have been developed: base editing and prime editing [41,42,43]. Both methods exploit catalytically nuclease deficient Cas proteins (deadCas9—dCas9, and nickase Cas9—nCas9 for base editing and prime editing, respectively) that alter the target DNA sequence without inducing double strand breaks. In the base editing system, dCas9 is fused to DNA deaminase domains allowing C to T (by cytosine base editor, CBE) or A to G (by adenine base editor, ABE) substitution. The prime editing system consists of a nCas9 fused to reverse transcriptase which allows the insertions, deletions, and point mutations at specific loci. Prime editing can generate targeted insertion (up to 44 bp), deletion (up to 80 bp), and all types of point mutations efficiently and precisely. In addition, by providing two steps of hybridization, it results in much lower off-targets than the standard Cas9 method which requires only one hybridization [44].

The delivery of CRISPR/Cas reagents in plant cells is obtained by genetic transformation, for which a relevant issue is represented by the species- and cultivar-specific regeneration efficiency with some genotypes being extremely recalcitrant. Several protocols have been developed to overcome this problem. The use of virus-derived vectors for delivering sgRNAs represents a promising method for shortening and simplifying the transformation process [40]. Usually, transgenic plants overexpressing Cas9 are further transformed with these vectors through agroinfiltration and the virus expressing system directs the expression of viral genes. Thus, the infected plant cells become reservoirs for viral genes that can spread across cells or germline cells, via a systemic infection. This method could allow the generation of transgene-free edited varieties, especially using RNA viruses that do not integrate into the plant genome. However, the large Cas9 encoding sequence limits the application of such viral vectors and requires DNA-based systems like DNA viruses, that integrate into the genome and possess a lower cell-to-cell movement, or *Agrobacterium* [45]. A DNA-free system consisting of an in vitro assembled Cas9-gRNA ribonucleoproteins (RNPs) can be delivered into the host by conventional transformation techniques, like particle bombardment of immature embryos or PEG-mediated transformation of protoplasts. The use of RNPs does not require DNA transformation but improvements are needed to ameliorate the efficiency and the portability, as the system is limited to the plant species with established tissue cultures, and the identification of primary transformants because of the absence of selection markers [45,46].

## 3. Suitable Plant Species for Vaccine Production

An effective high yield production of vaccines in plants rises from the optimal combination of transformation protocols, transgene expression, regulatory elements, optimal control of post-translational processes (mainly glycosylation processes), and purification methods [47,48]. Many plant systems have been suggested for vaccine preparation, including cereals (corn, rice, and barley), legumes (soybean) and horticultures (tomato, lettuce, spinach, and carrot, used specifically for edible vaccines production) [48,49,50]. Nonetheless, *Nicotiana* species and alfalfa are the most preferred platforms for injectable vaccine production thanks to the inexpensive and high biomass, seed yield, and to the rapid scale up. Alfalfa gives the opportunity to accumulate high protein amounts in leaves, even if its use for animal feed poses concerns for the related feed chain contamination. The genus *Nicotiana*, specifically *N. tabacum* (cultivated tobacco) and *N. benthamiana* (Australian tobacco), is easy and fast to transform and grows quickly [8]. *N. benthamiana* is well suited to produce recombinant proteins in controlled conditions because it propagates transient expression vectors easily, while *N. tabacum* is preferred for large-scale production in open fields [51]. As for other plant systems, *Nicotiana* species possess transcriptional, translational, and post-translational mechanisms like the ones displayed by mammalian cells and necessary for the function of many biopharmaceuticals [52]. Moreover, representing phylogenetically distant species from humans, they do not need control tests for animal pathogens during growth, contrary to in vitro mammalian cell cultures [31] and, being non-food/non-feed plants, they imply low risk of contamination of the materials of GM plants in the human/animal food chain [53,54].

Even though plant and mammalian translational processes are similar, the production of recombinant proteins in plants is still affected by the differential codon usage among species that can cause a strong reduction of the expression, in the host, of exogenous genes from other organisms [17]. Several methods to optimize the codon usage, based on mRNA sequence modification, have been developed to enhance the expression of recombinant vaccines/monoclonals in *Nicotiana* species [55,56,57,58].

Additionally, refinements for producing recombinant proteins with mammalian glycosylation in *Nicotiana* have been developed including the incorporation of human type N and O glycosylation pathways [59,60]. In parallel, CRISPR/Cas9 has been applied to inhibit the plant glycosylation pathway, allowing the expression of antibodies and/or recombinant proteins without plant-specific glycans which can greatly affect the immunogenicity, allergenicity, or activity of the proteins. To this aim, the genes responsible for plant-specific glycosylation, β(1,2)-xylosyltransferase (*XylT*) and α(1,3)-fucosyltransferase (*FucT*), were inactivated by editing two *XylT* genes and four *FucTs* in *Nicotiana tabacum* [15] and *N. benthamiana* [16]. The knock-out lines were then transformed with genes encoding for the human monoclonal antibodies and the purified antibodies did not display any β(1,2)-xylose or α(1,3)-fucose [15,16].

The optimization of codon usage and glycosylation pathway, the ease and rapidity of transformation, the fast growth, and the lower implication of safety concerns, with respect to other plant systems, make tobacco one of the most utilized hosts for plant factory systems [8,61]. Numerous examples of vaccines produced in *N. tabacum* and *N. nicotiana* are described in literature. Among these, vaccines against influenza viruses H5N1, HAI-05, and H1N1 produced in *N. benthamiana* and an edible vaccine to treat the hepatitis B virus expressed in tobacco are under clinical trials [62,63,64,65]. More recently, the VP40 antigen of Zaire ebolavirus was produced in *N. tobacum* and maintained its antigenicity and the ability to induce immune response in mice [66]; a candidate therapeutic vaccine against papillomavirus, consisting of the oncoprotein E7 fused with a bacterial cell-penetrating peptide, was expressed in *N. benthamiana* [67]; a synthetic gene expressed in tobacco and encoding for a capsid protein of the food and mouth disease virus was able to stimulate the immunogenic response in rabbits [56]; a recombinant vaccine for classical swine fever virus was produced at a cost-effective large scale in *N. benthamiana* [57]; a full-length hepatitis C virus glycoprotein E2, correctly processed and folded, was expressed in *N. benthamiana* and induced immune response in vaccinated mice [68].

Draft genomes of three *N. tabacum* varieties were released in 2014 [69] and, more recently, Edwards et al. [70] produced an improved genome assembly. A draft genome sequence is available even for *N. benthamiana* [71]. The genomes are available at https://solgenomics.net/ (accessed on 20 April 2021). Even if improvements to the assembly of these complex genomes are necessary, these resources provide genomic roadmaps helping the use of the two species as biofactories. Annotated genomes, in fact, are fundamental for the development of genome editing approaches, allowing a precise construction of sgRNAs to be used for an exactly directed gene targeting with minimal to no off-target effects.

Protocols for genome editing in *Nicotiana* species are well established [72,73,74]. The only drawbacks related to the exploitation of tobacco for biopharmaceutical production is the presence of toxin compounds like nicotine, which need to be eliminated during the extraction processes [48,50]. A solution to this problem was made with the application of CRISPR-Cas9 to inhibit nicotine biosynthesis [75].

## 4. Virus-like Particles (VLPs) as Best Candidates for Vaccine Production in Plants

In addition to recombinant antigens and monoclonal antibodies, which represent powerful tools to contrast infectious diseases, virus-like particles (VLPs) are of ever-increasing interest. VLPs are self-assembled nanoparticles resembling the molecular and morphological features of authentic viruses as being non-infectious and genomeless multiprotein structures. They consist of multiple highly ordered coat proteins (CPs), which mimic the natural conformation of viral proteins, without the ability to replicate due to the lack of genetic materials [76].

VLPs represent natural vaccine adjuvants. CPs, in fact, are strong mammalian antigens able to activate B cell responses [77], and the optimal small size, shape, and rigidity of VLPs allow a fast transport to the lymphatic tissues where the antigen-presenting cells (APCs) and humoral and cellular responses can be stimulated [78,79,80,81]. In addition, VLPs are advantageous when the antigen structure is complex and cannot be fully produced with peptide-vaccines, and do not require accessory proteins for budding from animal/plant cells thanks to their self-assembling capacity [82]. VLPs were also reported to trigger a higher immune response compared to recombinant soluble proteins [83,84] or inactivated viruses [79].

Vaccines based on recombinant VLPs (Chimeric Virus Particles (CVPs)) [78], exposing correctly folded antigens chemically or physically associated to the envelope, have been developed, and CVPs exposing multiple copies of antigens and immunostimulant components (i.e., T-cell epitopes) have been proposed as activators of long-lasting immunity [85,86,87,88,89,90]. Furthermore, the presence of epitopes on the CVP envelop speeds up and facilitates the purification procedure, avoiding the tedious use of affinity chromatography and SDS-PAGE [78].

The most common VLP-based vaccines are built on animal virus backbones, even though animal viruses are associated with some safety concerns. Plant VLPs, instead, represent a valid alternative since plant viruses are not able to infect mammalian cells and have high flexibility in the structural components that can be easily chemically or genetically modified [86]. Thanks to these properties, to the ease of VLPs production and purification, the high stability, and the low risk of pre-existing immunity, universal vaccine platforms have been developed [91,92,93,94,95] and many examples of plant VLPs-based human vaccines are reported in literature and summarized by Balke and Zentils [96]. The authors listed at least 71 experimental vaccines, 16 anti-cancer vaccines, and 10 vaccines against allergies, autoimmune, and neurodegenerative diseases. The review also describes the main plant viruses utilized for vaccine production which, usually, are non-enveloped or naked viruses, and made of genetic material included in CPs only [96].

The development of a CVP-based vaccine plant-based platform requires the assembly of an expression cassette that is transiently or permanently introduced in the plant cells via *Agrobacterium*. The expression cassette contains a promoter region of viral or bacterial origin, the viral sequences for replication, and the coding sequences for CP fused with the selected antigen [86].

A promising plant CVP-based vaccine is represented by Pfs25 VLP-FhCMB which is based on a chimeric non-enveloped VLP consisting of the *Plasmodium falciparum* antigen Pfs25 fused to the N-terminus of the Alfalfa Mosaic Virus (AlMV) CP, produced in *N. benthamiana* using a TMV (tobacco mosaic virus)-based hybrid expression vector. The CVPs have been then purified to demonstrate scalability and industrial production feasibility [97]. The efficacy of this vaccine has been demonstrated by an in-human phase I clinical trial [98]. More recently, a successful phase I trial was also conducted to assess the safety and immunogenicity of a novel plant VLP based on PapMV (Papaya Mosaic Virus) combined with trivalent influenza vaccine [99].

Currently, global efforts are concentrated in developing vaccines against the new pandemic SARS-CoV-2 virus. SARS-Cov-2 virions consist of four structural proteins, namely S, small envelope (E), membrane (M), and nucleocapsid [100,101,102]. All four structural proteins can elicit strong humoral and CD4+/CD8+ T-cell responses [103,104], even if the S protein is the most efficient in terms of antibody-based detection [102,105]. The protein represents the main candidate for vaccine production and at least three companies are expressing it in plants: Diamante (Verona, Italy; https://www.diamante.tech/, accessed on 20 April 2021) [105], Kentucky Bioprocessing (Owensboro, KT, USA; British American Tobacco, 2020; https://www.bat.com/, accessed on 20 April 2021), and Medicago Inc. (Quebec City, Canada; https://www.medicago.com/, accessed on 20 April 2021). A CPMV (cowpea mosaic virus)-based VLP for SARS-CoV-2 is under development at the John Innes Centre (JIC, Norwich, UK; https://www.jic.ac.uk/, accessed on 20 April 2021) as a diagnostic control reagent for the RNA-based assay for virus detection screenings [105]. The approach utilized is based on the one set up for the foot and mouth disease virus and consists in the production, in plants, of CPMV VLPs carrying artificial RNA encoding for the whole SARS-CoV-2 genomic regions detected by the screening kit WHO (World Health Organization) [105]. These experiments open the basis for plant-based production of vaccines and screening reagents to contrast the new pandemic, COVID-19, and related SARS strains.

## 5. Taking Advantage of Pre-Existing VPL Structures in the Plant Genome

Endogenous Pararetriviral Elements (EPREs) derived from reverse-transcribing DNA viruses (Pararetroviruses) are widespread in plant genomes [106,107,108,109,110,111]. EPREs correspond to entire viral genomes and usually are not associated with any disease but can maintain the replication competence, generating infection in specific hosts in certain environmental conditions [112]. This activation results in the assembly of virus particles [113]. Roles in the defense against infection by the cognate exogenous virus and in plant genome evolution and plasticity have been proposed for these viral integrated DNAs [107,112,114,115,116].

A genus of *Caulimoviridae*, the *Florendovirus*, has been discovered in plant taxa, from algae to flowering plants, contributing to more than 0.5% of total genome content [116,117]. *Partitivirus* CP-like sequences were identified in a wide range of plant species and the nucleocapsid protein genes of *Cytorhabdoviruses* and *Varicosaviruses* were found in species of over nine plant families, including *Brassicaceae* and *Solanaceae* [109]. In addition, DNA sequences related to *Geminiviruses*, having single stranded DNA genomes, were found integrated in various *Nicotiana* species [118].

Lockhart et al. [106] found a high level of similarity (from 73% to 92% for the four ORFs included in the viral genome) between the Turnip vein-clearing virus (TVCV) genome (AF190123), and a hypothetical viral genome assembled from *Pararetrovirus*-like sequences integrated in high copy number in the *N. tabacum* genome (TPV; AJ238747). To be noted, expression was discovered for TPVs related to TVCV in *N. tabacum* [113] and transition from latency via episomes was detected for TVCV EPRE in *Nicotiana edwardsonii* [106,112].

TVCV-derived CVPs produced in plants have been used in biopharmaceuticals for the isolation of immunoglobulins at high purity [119]. The C-terminal of CP of TVCV was fused to a functional fragment of the protein A of *Staphylococcus aureus* (an antibody-binding agent used for IgG purification), introducing a 15 amino acid linker or a helical linker peptide, and the construct was utilized to transform *N. benthamiana*. It has been demonstrated that monoclonal antibodies expressed in plants could be isolated by purified viral particles displaying functional protein A and crude extracts containing the CVP. This experiment not only demonstrates that TVCV sequences can generate functional VLPs, but also provides evidence of the possibility to successfully assemble virions displaying large protein fragments, and a simple protocol to purify antibodies from plant extracts [119]. This CVP was patented in 2009 [120].

BLAST searches using the TVCV genome assembled from *Pararetrovirus*-like sequences integrated in the *N. tabacum* genome (AJ238747) [106] detected 1324 and 147 highly significant homolog loci in the *N. tabacum* and *N. benthamiana* genomes, respectively (Appendix A). Figure 2 illustrates the organization and sequence similarities of TVCV genome and Ntab-TN90_AYMY-SS16611, one of the most significant TPV loci identified by the BLAST search on the *N. tabacum* genome.

All four TVCV ORFs show a high level of sequence similarity with the corresponding tobacco sequences. Accordingly, the predicted proteins encoded by the tobacco TVCV-related sequences were similar to the corresponding viral proteins and conserved the domains identified by Pfam searches (Figure 2). Protein secondary structures of CPs from TVCV and TPV were predicted and compared by Jalview [121] (Figure 3).

The 3D modelling was conducted by trRosetta [122] and visualized by RasMol [123] (Figure 4).

In general, the secondary structure of the TVCV CP is highly conserved in the tobacco CP (Ntab CP) even if the 3D prediction revealed differences in the conformations of the two proteins (Figure 4). The conservation of the secondary structures and the protein domains in the tobacco predicted proteins suggests that protein functionality can be maintained and the ability to generate functional VLPs cannot be excluded.

TVCV-related TPVs represent good candidates as target sites for genome editing aimed at producing a standardized host tobacco plant expressing functional CVPs. We hypothesized that few “corrections” at the natural existing TVCV-like sequences in the tobacco genome can result in the production of functional CVPs with minimal modification of the plant genome. The prime editing method can insert small DNA sequences up to 44 bp at a specific locus. Considering that the coding sequence for an epitope (10–15 amino acids) is usually shorter than 44 bp, prime editing might be applied to fuse the tobacco TVCV-related CP encoding sequence with the one corresponding to a selected epitope (Figure 5).

A small linker, as suggested by Werner et al. [120] could be also inserted. The obtained CP–epitope sequence might represent the target site of prime editing and/or CRISPR/Cas9 aimed at replacing the epitope encoding sequence for the generation of different CVPs specific for different strains or pathogens (Figure 5).

Another important issue to consider is the expression level of the fusion proteins. The CRISPR-Cas9 method could be applied to insert a constitutive strong promoter region (i.e., maize ubiquitin) by designing a sgRNA specific for the sequence just upstream of the tobacco TVCV-related CP locus (Figure 5). Moreover, using the CRISPR/Cas techniques for the regulation of gene expression, CRISPR activation (CRISPRa) or CRISPR-based repression (i.e., CRISPR interference - CRISPRi), it is possible, thanks to their RNA-guided nature, to specifically control the expression of target genes more precisely and efficiently with respect to conventional methods [124,125,126,127].

Further studies are needed to evaluate the ability of tobacco TVCV-related CPs to generate functional VLPs. However, CRISPR/Cas9 might be also utilized to replace the tobacco CP sequence with the TVCV-specific ORF1, encoding for TVCV CP (Figure 5). Such application would be based on the activity of HDR, thus the recent improvements aimed at increasing the occurrence of HDR should also be applied.

This strategy depicts a hypothetical experiment where genome editing techniques take advantage of the pre-existing sequences from tobacco genome, overcoming the difficulty in genetic manipulation of large full-length or near-full-length viral genomes and the tedious procedures to integrate this structure in the plant genome. The development of a standardized plant host, easily modifiable for specific CVP expression, would be useful in the case of pandemics when it is necessary to rapidly contrast the occurrence of new strains/pathogens.

## 6. Simplified Industrial Production of Vaccines in Plant by Combining VLPs and Genome Editing

The industrialized production of vaccines requires consistent and validated processes and control assays of each step to ensure that the final product matches the quality standards for human use. The main production steps and the relative quality control tests are summarized in Figure 6.

One of the most important steps in an industrialized production process is represented by purification. In the case of plant-based systems, the whole plant or plant tissues/organs are harvested and processed in purification facilities under current Good Manufacturing Practices (GMP). In the upstream processing a clarified extract is generated and moved to downstream processing and eventually formulated to produce the material according to quality standards. The recovery of recombinant proteins from host cells/tissues can be carried out by physical or chemical cell lysis. Centrifugation, depth filtration, microfiltration, or tangential flow filtration (TFF) are used for clarifying the cell lysate. Microfiltration is preferred in large production due to the robustness and scalability. An endonuclease treatment with Benzonase may be employed to degrade residual nucleic acids. The ultrafiltration/diafiltration/ultracentrifugation step may be included in the process to concentrate and buffer exchange the product, removing endonuclease, and also makes it ready for the next steps [128,129]. During concentration and buffer exchange the host cell proteins are further reduced. Purification by anion exchange chromatography is additionally used to decrease the amount of DNA and proteins from the host or eventual plant toxins. As an example, the affinity column purification process for recombinant therapeutic proteins expressed from tobacco leaves can remove alkaloids [130]. Altogether, the entire purification procedures represent the bottleneck for vaccine production as being the most time consuming, laborious, and expensive step. Moreover, each vaccine candidate demands specific procedures.

Plant-based systems are often described as cost-effective due to the low cost of upstream cultivation. Different groups tried to estimate manufacturing costs for plant-based processes for monoclonal antibodies or other biopharmaceuticals, and concluded that cultivation accounts for only a small part of the process costs, while the downstream processing, especially purification, represents the major weakness that limits the commercial utilization of plant-based biopharmaceuticals [30,131,132].

A possible solution is to reduce the costs of downstream processing, for example, reducing the chromatographic steps. The use of VLPs, which require a faster purification in comparison to recombinant proteins, could sensibly reduce the purification costs. During the recovery of VLPs, a treatment of crude extracts with detergent or heat is enough to achieve selective enrichment of VLPs and optimal treatment conditions can allow the removal of more than 90% of endogenous plant proteins without any loss of the product [133]. The size and structure of these complexes allow the application of a quick procedure based on ultracentrifugation on Cesium Chloride or Sucrose gradient, or isopycnic centrifugation, accelerating the purification steps and reducing the need of purification equipment [129].

The purification process is designed to remove host toxins, noxious metabolites, and host cell proteins to an acceptable level in compliance with the pre-established quality control specifications [4]. The time and costs related to the elimination of toxic compounds can be reduced, achieving simplification of the purification process by the targeted silencing of genes implicated in their biosynthesis in the host, as recently proposed by Buyel et al. [17]. As mentioned above, an example of this application is represented by the deployment of CRISPR/Cas9 to inhibit nicotine biosynthesis in *N. tabacum* [50]. The reduction of host cell proteins might be obtained, addressing genome editing to reduce the expression of the most abundant proteins, such as the ones involved in photosynthesis or seed storage proteins [17].

Despite the advances geared towards purifying large-scale amounts, technical challenges persist in ensuring no/little contamination by host DNA is present in the vaccine product. To this end, CRISPR/Cas9 might help in eliminating the need of selection makers whose presence in the final product need to be thoroughly checked [134].

Thus, by combining the plant VLP platform with CRISPR/Cas9, the purification process can be simplified to achieve good yields in a faster and cheaper way.

The assessment of manufacturing consistency includes the evaluation of critical quality parameters and their corresponding attributes: identity, purity, safety, and characteristics (Figure 5).

After production and control tests, the uniformity of the products must be assured by validating batches and checking their similarity and, where possible, the consistent antibody response to the produced antigens in preclinical models or during clinical development. Nonetheless, for plant-based systems, the lack of the consistency of transgene expression in different batches and individual plants within the same batch has always represented a barrier for the successful application of the plant-based vaccines. Big improvements on lot-to-lot consistency have been obtained since plant-based vaccines expression technology was introduced almost two decades ago, using optimized conditions for *N. benthamiana* plants subjected to transient transformation [5,135,136]. Only recently, proof of lot-to-lot consistency of a plant-derived vaccine has been achieved in a clinical trial: three sequential lots of a quadrivalent virus-like particle influenza vaccine (QVLP) produced in *N. benthamiana* elicited equivalent antibody responses to the targeted hemagglutinin (HA) proteins in a phase III clinical trial. This analysis showed for the first time that a plant produced vaccine can meet the standard criteria for consistency of production [6].

CRISPR/Cas9, offering the possibility to integrate exogenous DNA in specific sites in a genome and to control the spatial/temporal gene expression, will help further to overcome these consistency issues and, with easy refinements of the production processes, the quality and fulfilment of the requirements of regulatory agencies (i.e., compliance with the International Council for Harmonisation of Technical Requirements for Pharmaceuticals for Human Use—ICH guidelines) could be easily achieved.

## 7. Other Advantages of Plant-Based Vaccine Production

One of the main issues related to vaccine production is the need of adjuvants to improve vaccine efficacy. Even in the case of the self-adjuvanting VLPs, all licensed VLP-based vaccines are currently formulated with aluminum salts or other adjuvants [137]. Since some plant components [138] have adjuvating ability, plant cell encapsulated VLPs may not need additional adjuvants, further reducing the costs of production. Lyophilized tissue, indeed, represents a delivery system that may be cost-effective, foregoing both VLP purification and potentially the use of additional adjuvants, as well as requiring less storage capacity. Additionally, antigen or VLP structure and immunogenicity is preserved when plant tissue is lyophilized, and it has been shown to be stable for at least one year [139]. The stability can also be enhanced, however, by adding excipients and stabilizing compounds [140].

Vaccines expressed in plants also have the benefit of no risk associated with the host DNA carried in a vaccine dose, given the phylogenetic distance between plant and human [6]. Thus, all the production steps aimed at minimizing the risk of host cell nucleic acid oncogenicity (i.e., reduction of DNA size to 100–200 base pairs in length), required for vaccines expressed in mammalian cell lines [141], can be reduced, decreasing the time, costs, and quality control tests. Overall, for plant systems, the steps to ensure purity are fewer and less stringent with respect to the ones used to ensure comparable quality of vaccines expressed in other platforms. For example, validated molecular tests for bovine viruses and cell culture tests of bovine sera—freedom from phage, endotoxin, or oncogenes—are strictly regimented and tested in biotherapeutics expressed in mammalian or yeast platforms. The source(s) of any component of animal origin for the bovine spongiform is regulated for encephalopathy risk. Additionally, when an insect or mammalian cell substrate is used to produce the biotherapeutic, the production process should be validated for its capacity to eliminate (by removal and/or inactivation) adventitious viruses [142]. By this view, plants result in less expensive and safer platforms to produce vaccines and antibodies compared to the other “traditional” systems.

## 8. Conclusions and Future Perspectives

The COVID-19 pandemic has created a tremendous rise in demand for vaccines, highlighting the gaps in the capacity to rapidly produce biopharmaceuticals in emergency situations [8]. The plant factory system can address some of the most growing health concerns worldwide and can provide a fast response to epidemics/pandemics. Plant-based platforms have numerous advantages in vaccine production given their scalability and genome plasticity. Furthermore, plant systems allow obtaining safe products with a less complicated production process and quality controls compared to mammalian and bacterial production platforms.

The expression of candidate vaccines in plants exploits their eukaryotic processing machinery, supporting appropriate post-translational modifications and assembly of antigens. Additionally, plant-derived VLPs may have a significant safety advantage since the risk of contamination with human pathogens is extremely low, allowing a simplification of purification and control steps.

Recombinant proteins can be easily expressed in plants after transient or stable transformation of the host. Even if transient transformation offers a fast protein expression, with high flexibility and scalability, the stable transformation has been considered the most promising technique for plant vaccines production. The introduction of the CRISPR/Cas9 technique has faced regulatory bodies unprecedented times [143], providing the generation of well characterized, highly controlled, and off-targets free GM plants. Additionally, it can allow the generation of transgene-free plants that do not contain foreign DNA as GM organisms, and null segregants can be obtained. This could lead to the possible development of a standardized plant host where transgene-free plants can be generated with the deletion of unwanted secondary metabolites or toxins. In this way, it would be possible to produce master seed banks of a such standardized plant host satisfying industrial needs, and simplifying the containment measures, the related environmental risk assessments, and facilitating regulatory requirements [144,145,146].

Endogenous genomic loci of viral origin, reminiscent of viral integrations, represent useful sites for the insertion of sequences encoding for VLPs displaying the antigenic protein. In a likely oncoming scenario, this can allow minimal targeted mutagenesis of the transgene-free standardized plant host, expediting the release and commercialization of the products of genome edited plants.

In addition to the control of off-targets and integration site, CRISPR/Cas9 also gives a solution for the presence of selection markers. Thus, once the seed banking system including assessment of the genetic stability are documented, and specifications for qualifying seeds and data (including information on manufacturing process that must comply with current GMP, the quality attributes of the resulting product, and safety and biological activity of the product) produced, specific classified seeds/plants can be generated and the site of manufacture could be possibly licensed for contained use and production processes in a similar way to a conventional biotechnology production facility (in Europe: Contained Use’ Directive 2009/41EC). Once this is achieved, vaccines could be produced with unprecedented time and yield compared to other platforms.

## Figures and Tables

**Figure 1 plants-10-01828-f001:**
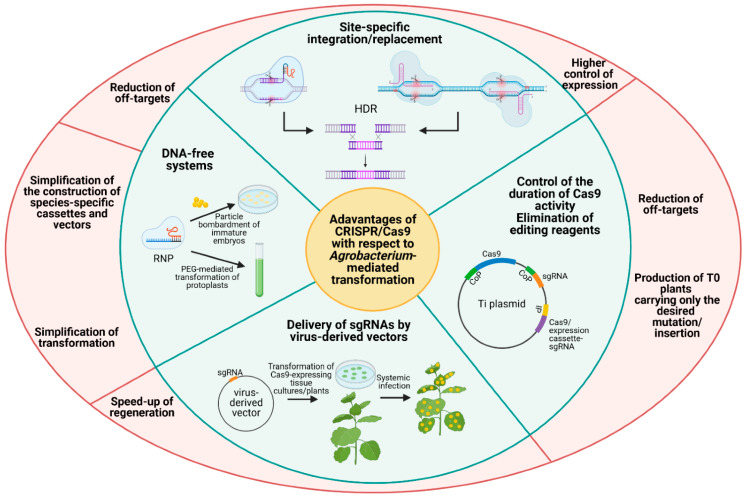
Advantages of the CRISPR/Cas9 method with respect to *Agrobacterium*-mediated transformation. The peculiarities of CRISPR/Cas9 (indicated in the green part) with the corresponding ameliorations (indicated in the red part), in comparison to the conventional *Agrobacterium*-based transformation, are represented. HDR = homologous-derived repair, CoP = constitutive promoter, sgRNA = single guide RNA, RNP = ribonucleoprotein. Created with Biorender.com (https://biorender.com/; accessed on 20 April 2021).

**Figure 2 plants-10-01828-f002:**
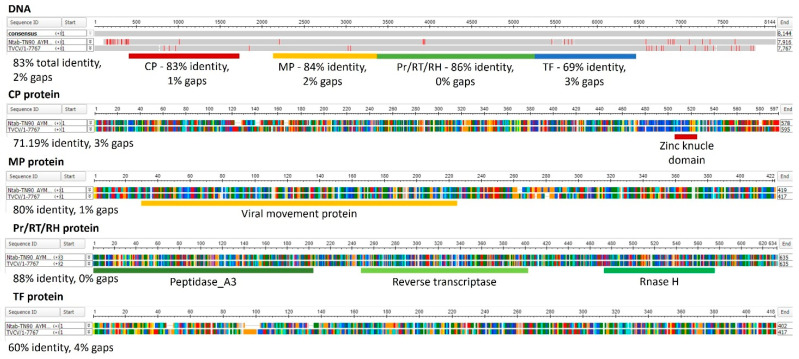
Alignments of tobacco TVCV-related sequence Ntab-TN90_AYMY-SS16611 and TVCV genomic sequences and the predicted proteins. Alignments were obtained by Clustal Omega (https://www.ebi.ac.uk/Tools/msa/clustalo; accessed on 20 April 2021) and visualized by Multiple Sequence Alignment Viewer 1.20.0 (https://www.ncbi.nlm.nih.gov/projects/msaviewer; accessed on 20 April 2021). The percentage of sequence similarity and gaps are reported for each alignment. The four ORFs in the TVCV genome are highlighted for the DNA alignment: red—CP (coat protein) encoding ORF1, yellow—MP (movement protein) encoding ORF2, green—Pr (peptidase)/RT (reverse transcriptase)/RH (Rnase H) encoding ORF3, blue—TF (transactivator factor) encoding ORF4. Red bars in the DNA alignment indicate sequence differences. The protein domain detected by Pfam (http://pfam.xfam.org/; accessed on 20 April 2021) are indicated in the protein alignments. The function of RasMol amino acid colors has been utilized.

**Figure 3 plants-10-01828-f003:**
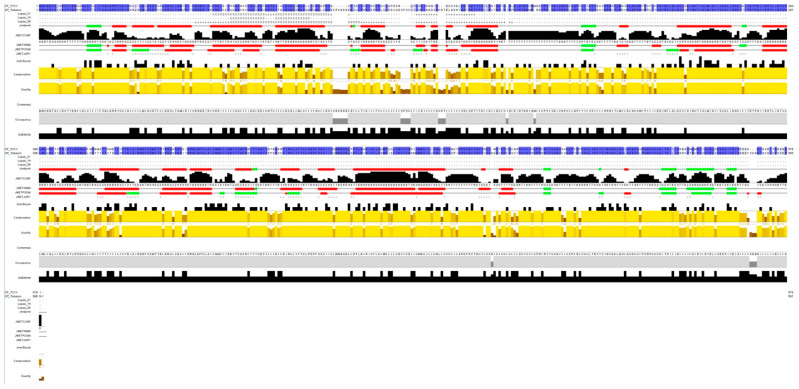
Jalview alignment of the CP encoded by Ntab-TN90_AYMY-SS16611 and TVCV. Conserved residues are indicated in blue in the Jalview alignment [121]. Conservation and quality of the alignment are indicated. Secondary structures, predicted by Jpred Secondary Structure Prediction are indicated in green (β-sheets) and red (α-helices); JNetCONF = the confidence estimate for the prediction; JNetHMM = HMM profile-based prediction; JNETPSSM = PSSM-based prediction; JNETJURY = a ‘*’ in this annotation indicates that the JNETJURY was invoked to rationalize significantly different primary predictions.

**Figure 4 plants-10-01828-f004:**
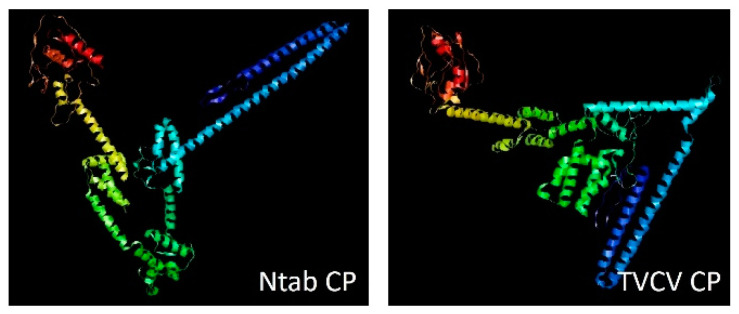
Predicted 3D structures of Ntab-TN90_AYMY-SS16611 and TVCV CPs. The 3D structures have been predicted by trRosetta [122] and visualized by RasMol [123] from the pdb files obtained by trRosetta [122].

**Figure 5 plants-10-01828-f005:**
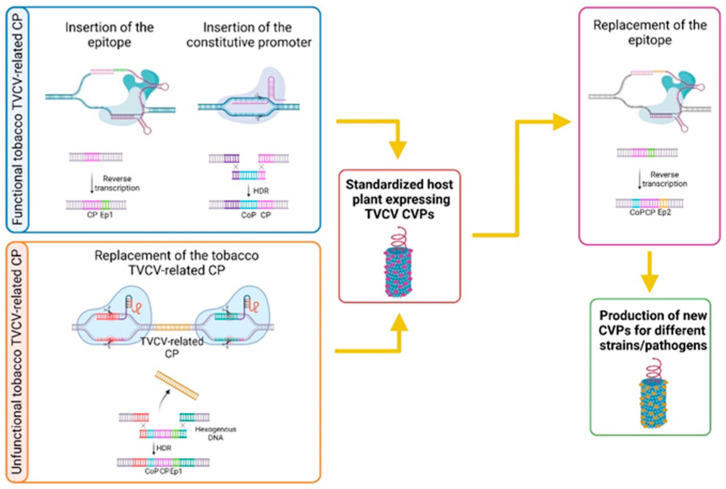
Steps for the generation of a standardized tobacco host for vaccine production through the association between CVP and genome editing techniques. Genome editing can be used to manipulate the TVCV-like sequences integrated in *N. tabacum* genome showing high sequence similarity to TVCV CP encoding ORF. Two possible strategies can be postulated depending on the ability of the tobacco CP corresponding sequences to generate VLPs. (1) Functional tobacco TVCV-related CPs: a small epitope encoding sequence can be fused to the tobacco CP-related ORF by prime editing; a constitutive promoter region can be integrated upstream the CP-related ORF by CRISPR-Cas9 taking advantage of HDR. (2) Unfunctional tobacco TVCV-related CP: the tobacco CP-related sequence could be replaced by the TCVC CP ORF carrying a constitutive promoter and the epitope encoding sequence by using CRISPR-Cas9 with sgRNAs targeting sites flanking the tobacco locus. The standardized host plant, able to produce functional CVPs, can be successively modified by replacing the epitope sequence through base editing in order to obtain functional CVPs towards different pathogens/strains. CP = coat protein, Ep1 = epitope 1, Ep2 = epitope 2, CoP = constitutive promoter. Created with Biorender.com (https://biorender.com/; accessed on 20 April 2021).

**Figure 6 plants-10-01828-f006:**
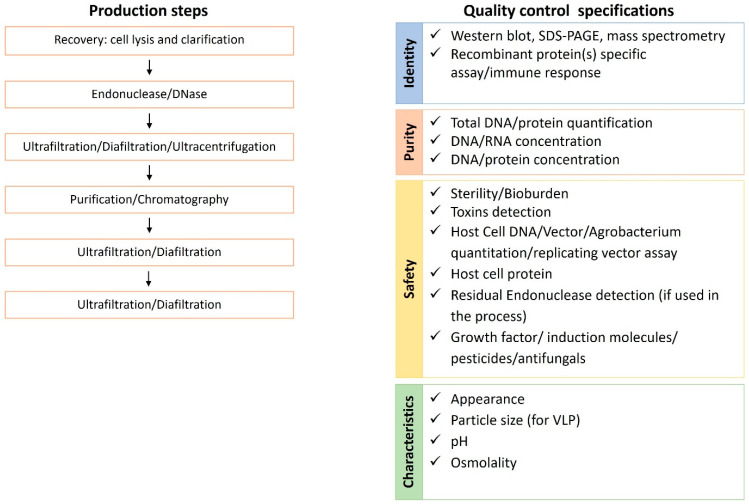
Summary of the production steps and quality control specification of industrial vaccines production.

## Data Availability

Not applicable.

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
