# Peer review of "Frontiers in the Standardization of the Plant Platform for High Scale Production of Vaccines"

_plants, 2021, doi:10.3390/plants10091828_

Round 1

Reviewer 1 Report

In this review paper, Citiulo et al. tried to highlight the potential of plant genetic engineering approaches for the production of vaccines. However, before reaching any decision, some concerns should be addressed:

1) what is the novelty of this review paper in comparison with khan et al. (2021) paper? (Khan T, Khan MA, Karam K, Ullah N, Mashwani ZUR and Nadhman A (2021) Plant in vitro Culture Technologies; A Promise Into Factories of Secondary Metabolites Against COVID-19. Front. Plant Sci. 12:610194. doi: 10.3389/fpls.2021.610194)

2) In some sections (e.g., 2. Transformation technologies and CRISPR-Cas genome editing methods, the approaches of genome editing and gene transformation have been presented. But it seems that this section is out of the scope of this review.

Importantly, there are some new technologies such as epigenome editing, CRISPRi, CRISPRa, etc., which are required for vaccine production.

3) Section 4 (Virus-Like Particles (VLPs) as best candidates for vaccine production in plants) has highlighted the potential of viruses as a promising particle for gene transformation. If the aim of this study is vaccine production in plants, why did the authors discuss transformation methods?

4) Generally, it is necessary to clarify the aim of this review paper. Is the aim of this study to present the plant genetic engineering methods for vaccine production?!? If yes, what is the novelty of this review paper in comparison with khan et al. (2021) paper? If no, please clarify the aim of this study.

Author Response

Reviewer 1

In this review paper, Citiulo et al. tried to highlight the potential of plant genetic engineering approaches for the production of vaccines. However, before reaching any decision, some concerns should be addressed:

  • what is the novelty of this review paper in comparison with khan et al. (2021) paper? (Khan T, Khan MA, Karam K, Ullah N, Mashwani ZUR and Nadhman A (2021) Plant in vitro Culture Technologies; A Promise Into Factories of Secondary Metabolites Against COVID-19. Front. Plant Sci. 12:610194. doi: 10.3389/fpls.2021.610194)

Authors’ reply: Our review work describes the most renewed techniques of vaccines development in plants (i.e., VLPs and industrialization of the production process) and the conjugation of these vaccine technologies with genome editing methods based on CRISPR/Cas.

We highlighted the importance of the possibility that plant endogenous genomic loci of viral origin, reminiscent of viral integrations, could represent useful sites for the insertion of sequences encoding for VLPs displaying the antigenic vaccine candidate, allowing minimal targeted mutagenesis of the transgene-free standardized plant host.

We commented, based on the most recent literature, that these progresses all together could result in the standardization of the plant platform with the overtaking of constancy of production challenges and facilitation of regulatory requirements, expediting the release and commercialization of the vaccine products of genome edited plants. We concluded that this could give an added value for the use of plant platforms for large-scale manufacture of vaccines.

The valuable work conducted by Khan et al. (2021) describes the important roles of plant secondary metabolites or phytochemical compounds against infectious disease taking coronaviruses as an example. Therefore, the authors commented on the value of the in vitro cultures that, with the aid of the plant biotechnology, can serve as factories of secondary metabolites/phytochemicals that can be produced in bulk and standardized quality (these compounds might help in the fight against COVID-19). They also commented on the fact that genetic manipulation of these in vitro cultures could provide engineered drug candidates.

We agree with Reviewer 1 that the progress in the genetic manipulation have an additional benefit in the industrialization of the plant factory (note in vitro culturing vs plant host) in terms of consistency and yield as indicated by Khan et al. (2021). Nonetheless, the in vitro culture-based phytochemicals research field and the vaccine field are distinct, and both use ad hoc techniques and production purification processes. Thus, we believe that both reviews are self-standing and present novelties of interest for the phytochemical/small molecule compounds and the vaccine research fields, respectively.

  • In some sections (e.g., 2. Transformation technologies and CRISPR-Cas genome editing methods, the approaches of genome editing and gene transformation have been presented. But it seems that this section is out of the scope of this review.

Authors’ reply: Section 2 (Transformation technologies and CRISPR-Cas genome editing methods) includes a brief description of the conventional plant transformation methods and a more detailed description of the CRISPR/Cas technologies, underlying the genome editing-related advantages, with respect to the classical transformation, that are revolutionizing plant biotechnology. As the aim of the present manuscript consists in the illustration of the potential conjugation of the renewed vaccine technologies (i.e., VLPs and industrialization of the production process) with genome editing to generate standardized plant hosts with the overtaking of constancy of large-scale production challenges, expediting the release and commercialization of the vaccine products of genome edited plants, we believe that Section 2 is important to give an overview of the advantages of the CRISPR/Cas-based genome editing and its potential uses in vaccine production.

Importantly, there are some new technologies such as epigenome editing, CRISPRi, CRISPRa, etc., which are required for vaccine production.

Authors’ reply: The technologies suggested by the Reviewer, with the corresponding citations (109-112) have been added in the text (from lane 418 to lane 422).

  • Section 4 (Virus-Like Particles (VLPs) as best candidates for vaccine production in plants) has highlighted the potential of viruses as a promising particle for gene transformation. If the aim of this study is vaccine production in plants, why did the authors discuss transformation methods?

Authors’ reply: VLPs represents valuable antigens carriers thanks to their properties, as described in the manuscript. One of the main points described in the review consists in the possibility that plant endogenous genomic loci of viral origin, reminiscent of viral integrations, could represent useful sites for the insertion of sequences encoding for VLPs displaying the antigenic vaccine candidate, allowing minimal targeted mutagenesis of the transgene-free standardized plant host.

Moreover, as indicated in the reply to comment 1, a brief description of the conventional transformation methods has been provided, together with the description of the genome editing methods, to highlight the advantages that the new genome editing methods have brought to plant biotechnology, with respect to conventional transformation. This agrees with the aim of the review that consists in providing an overview of a potential conjugation of the renewed vaccine technologies with genome editing to expedite the release and commercialization of vaccines produced with constancy at large scale in genome edited plants.

  • Generally, it is necessary to clarify the aim of this review paper. Is the aim of this study to present the plant genetic engineering methods for vaccine production?!? If yes, what is the novelty of this review paper in comparison with khan et al. (2021) paper? If no, please clarify the aim of this study.

Authors’ reply: The aim of the manuscript has been clarified in the Abstract (from lane 20 to lane 26) and in Introduction (from lane 73 to lane 78). The comparison to Khan et al. (2021) has been widely commented in Authors’ reply to Reviewer’s comment 1.

Reviewer 2 Report

This manuscript explained the vaccine production technologies with genome editing technique based on CRISPR/Cas9 method in plants. I believe this review might help to readers of PLANTS after minor revisions.

- Indicate the scientific name in italics.

  1. ex) N. benthamiana, N. tabacum, Staphylococcus aureus

- There are a number of omissions. Check carefully.

Fig. 1. Simplification of the construction of species-specific cassettes and vectors

Line 136. CRISPR-Cas9

Author Response

Reviewer 2

This manuscript explained the vaccine production technologies with genome editing technique based on CRISPR/Cas9 method in plants. I believe this review might help to readers of PLANTS after minor revisions.

- Indicate the scientific name in italics.

  1. ex) N. benthamiana, N. tabacum, Staphylococcus aureus

- There are a number of omissions. Check carefully.

Fig. 1. Simplification of the construction of species-specific cassettes and vectors

Line 136. CRISPR-Cas9

Authors’ reply: The whole manuscript has been revised according to Reviewer’s suggestions.

Reviewer 3 Report

Review - Paper Plants MDPI

Title: Frontiers in the standardization of the plant platform for high 2 scale production of vaccines

The aim of this manuscript is to expose the advantages and methods of vaccine production in plants. Currently, any information about vaccines is a “hot” topic. The manuscript attended its purpose and presented very interesting information about the subject. However, there are some changes that have to be made.

The authors focused a lot on plant transformation methods and CRISPR, I believe they could give more emphasis on vaccines that are already being produced in plants. There is not much information about it. 

English needs to be proofread.

Lines 20-23: rephrase the sentence.

Line 44: remove the word “sponsored”. It appears twice in the same sentence.

Line 52: use the term “in planta” in italic

Lines 61-62: repeated from abstract

Lines 64-65: repeated from abstract

Line 75: use the terms “in vivo ” and “in vitro” in italic

Figure 1: It does not help in any clarification. A figure showing a comparison of the advantages and disadvantages of plant transformation methods would be better for this purpose.

Paragraphs starting in line 214 and ending in line 226 should be removed. Does not fit in the subject. Instead more studies showing vaccine production in plant species should be presented.

Line 242: correct “trig” to “trigger”

Line 475 Should it be there? It seems to not make sense.

Add information or more about:

  • The codon usage for species/kingdom is a topic that should be addressed.
  • The emphasis on the advantages of using a eucaryote, that is phylogenetically distant from humans, but still presents a very similar transcription and translation machinery should be a high point in this review.
  • What would be the risk of producing vaccines in animal and human food-plant species?
  • Describe better the methods for recombinant proteins extraction and purification from plant tissue, comparing time consumed, costs, and other pros/cons with bacterial production.

Author Response

Reviewer 3

The aim of this manuscript is to expose the advantages and methods of vaccine production in plants. Currently, any information about vaccines is a “hot” topic. The manuscript attended its purpose and presented very interesting information about the subject. However, there are some changes that have to be made.

The authors focused a lot on plant transformation methods and CRISPR, I believe they could give more emphasis on vaccines that are already being produced in plants. There is not much information about it. 

Authors’ reply: All Reviewer’s observations have been considered and all the suggested changes have been made. Please find Authors’ replies to the following comments.

English needs to be proofread.

Authors’ reply: The manuscript has been revised according to Reviewer’s suggestions.

Lines 20-23: rephrase the sentence.

Line 44: remove the word “sponsored”. It appears twice in the same sentence.

Line 52: use the term “in planta” in italic

Lines 61-62: repeated from abstract

Lines 64-65: repeated from abstract

Line 75: use the terms “in vivo” and “in vitro” in italic

Figure 1: It does not help in any clarification. A figure showing a comparison of the advantages and disadvantages of plant transformation methods would be better for this purpose.

Authors’ reply: Figure 1 aims to represent the advantages of CRISPR-Cas with respect to the conventional transformation methods mainly based on the use of Agrobacterium. We think that the Figure represents an important part of the review, highlighting, as well, the novelty of the work and visually concretizing in the mind of the reader important concepts that are although extensively commented in the main text. For a better elucidation of the Figure, the caption has been improved.

Paragraphs starting in line 214 and ending in line 226 should be removed. Does not fit in the subject. Instead more studies showing vaccine production in plant species should be presented.

Authors’ reply: The manuscript illustrates the potential synergy between the renewed vaccine technologies (i.e., VLPs and industrialization of the production process) and genome editing to generate standardized plant hosts with the overtaking of constancy of large-scale production challenges, expediting the release and commercialization of the vaccine products. We highlighted the possibility that plant endogenous loci of viral origin, reminiscent of viral integrations, could represent useful sites for the insertion of sequences encoding for VLPs displaying the antigenic vaccine candidate, allowing minimal targeted mutagenesis of the transgene-free standardized plant host. To this aim we proposed a potential application based on the use of pararetroviral elements in N. tabacum genome. We believe that the Paragraph indicated by the Reviewer is an important part of the review, providing important information about the possibility to apply CRISPR/Cas-based genome editing for vaccine production in plants like N. tabacum/N. nicotiana for which genome information are available and genome editing procedures are well established.

As recommended by the Reviewer, we presented additional studies of vaccines produced in plants, focusing on the ones expressed in Nicotiana species (from lane 241 to lane 253) that represent optimal hosts for recombinant protein expression as largely discussed in paragraph 3 (3. Suitable plant species for vaccine production).

Line 242: correct “trig” to “trigger”

Line 475 Should it be there? It seems to not make sense.

 Authors’ reply: The manuscript has been revised according to Reviewer’s suggestions.

Add information or more about:

  • The codon usage for species/kingdom is a topic that should be addressed.

Authors’ reply: the topic has been discussed (from lane 221 to lane 226).

  • The emphasis on the advantages of using a eucaryote, that is phylogenetically distant from humans, but still presents a very similar transcription and translation machinery should be a high point in this review.

Authors’ reply: the topic has been discussed (from lane 214 to lane 218).

  • What would be the risk of producing vaccines in animal and human food-plant species?

Authors’ reply: the advantage of producing vaccines in non-food/non-feed plants has been better explained (lanes 219 and 220).

  • Describe better the methods for recombinant proteins extraction and purification from plant tissue, comparing time consumed, costs, and other pros/cons with bacterial production.

Authors’ reply: From lane 501 to lane 506, we added considerations on manufacturing costs for plant-based processes, highlighting that reducing the costs of downstream processing may increase the utilization of the plant-platforms to produce biopharmaceuticals. Specifically, we suggested that the purification costs for vaccine production in plants may be reduced using VLPs (from lane 509 to lane 517).

Round 2

Reviewer 1 Report

All the comments have been addressed. I believe that this version of the manuscript can be published in Plants.